# The Impact of Education and Residential Environment on Long-Term Waste Management Behavior in the Context of Sustainability

**Andreea Simona Saseanu [1], Rodica-Manuela Gogonea [2,\*], Simona Ioana Ghita [2,3] and Radu Şerban Zaharia [4,\*]**

[1] Department of Business, Consumer Sciences and Quality Management,
The Buchares University of Economic Studies, 010374 București, Romania

[2] Department of Statistics and Econometrics, The Bucharest University of Economic Studies,
010552 Bucharest, Romania

[3] Institute of National Economy, 050711 Bucharest, Romania

[4] Ph.D School, VALAHIA University of Targoviste, 13000 Targoviste, Romania

\* Correspondence: manuela.gogonea@csie.ase.ro or manuela.gogonea@gmail.com (R.-M.G.);
rzaharia992@gmail.com (R.Ş.Z.); Tel.: +40-724-742-477 (R.-M.G.)

**Abstract:** Currently, the problem of waste reduction is a permanent concern for all countries of the world, given the need to ensure the sustainability development. In this context, the research aims to highlight the impact of education and demographic factors by residence areas on the long-term behavior of the amount of waste generated in 29 European countries during 2013–2017. The study is based on statistical and econometric modeling aimed at identifying, testing and analyzing the existence of long-term correlation between the amount of waste per capita recorded in each country and four factors of influence considered significant for waste reduction: *Pupils and students by education level* and *Classroom teachers and academic staff by education level*, representing exogenous variables which quantify the educational outcomes, as well as *The population by degree of urbanization* (cities, rural areas), as demographic factors. As a result of an analysis based on correlation and regression method, a cointegration relationship between the analyzed variables was identified. Considering the amount of waste as an important component of the environmental pressure, the obtained results show the significant long-term effect that education and the demographic factor can have on its long-lasting behavior, as well as the ways through which these factors can act to strengthen sustainability.

**Keywords:** sustainability; waste; education; degree of urbanization; cointegration model; Vector Error Correction models; autoregressive vector patterns

## 1. Introduction

Worldwide and implicitly at the European level, the issue of waste reduction is increasingly being addressed in the context of sustainable economic and social development of any country. This aspect takes into account the fact that sustainability is an integral component in the development strategy of each individual company and society as a whole, because it manifests itself in the interrelated plan between the three parts: economic, social and environmental. Closely interdependent with the environment, the evolution of waste is also closely linked to the strategic policy of preventing its generation and favoring recycling, so that the decision of prescriptive regulations or educational campaigns could be taken [1].

The concept of "waste" is oriented, on the one hand, to products that are at the end of the production and utility life cycle, and on the other hand to construction and demolition, production and trade

processes. Manufacturing each product includes a significant amount of waste during the life cycle, highlighting all the impacts and types of pressures exerted: thus, the impact on soil degradation, emissions, etc. occurs in the procurement of raw materials; the impact of production is signaled through the use of materials and energy, air and water emissions, solid waste, etc., and the impact of transport—through raw materials and products directed to the final markets [2].

The education of young people in the field of environmental protection both in family and in schools is a priority, representing an efficient way to reduce the amount of waste [3]. In this direction, studies have been carried out [4–6] certifying the importance of education and training in the field of environmental protection, raising public awareness of environmental responsibility, minimally invasive environmental lifestyles based on recycling and waste reduction, essential components for sustainability.

Liu and Wu [7] identified three main components in the waste management process: urban economy and urban development (where the urban population has the highest share), energy consumption and urban scale. At the same time, other important studies have shown that the amount of waste is directly influenced by the residential environment of households, with significant differences according to the location of the household (in rural or urban areas) [8–10].

In this context, the present research was designed to find out whether there is a long-standing relationship between the amount of waste in each country—as an endogenous variable—and the exogenous variables represented by the population of each country by urbanization degree (cities, towns and suburbs, rural areas), pupils and students by education level and classroom teachers and academic staff by education level, respectively, thus representing a strategic point of using and modeling the evolution of the waste amount, in the direction of its diminishing.

Taking into account that education on waste management starts from the first years of life, inside the family, and continues to be enhanced with each level of school education, two variables have been introduced into the study reflecting how education in pre-university and university field influences the fluctuation of the amount of waste in the direction of its reduction: Pupils and students by educational level and Classroom teachers and academic staff by education level. At the same time, the existence of significant differences in the behavior of generated waste in relation to the location of the rural or urban household has led to the need to consider the population by degree of urbanization (cities, rural areas) as exogenous variables.

The paper is organized in six sections. The first section—Introduction—presents briefly aspects of the context of addressing the impact of education and demographic factors by residential areas on the long-term waste behavior, in the context of sustainability. This was completed by the second section, which includes the literature with the presentation of many results of studies, through which several points of view of both the interested persons and the specialists dealing with waste management were highlighted in the context of the sustainability phenomenon. Section 3 describes the methodology used to identify the influence of the level of education and the degree of urbanism on the long-term waste evolution, a methodology that includes building and validating cointegration models.

The analysis—whose results are presented in Section 4—is based on statistical data on waste evolution, population by degree of urbanisation (cities, rural areas), pupils and students, respectively classroom teachers and academic staff by education level, in the EU states and Iceland.

The paper finally includes two sections (Discussion and Conclusions) summarizing the results of the analysis and outlining aspects of managerial possibilities and proposals regarding the impact of factors in reducing the amount of waste in the future.

The main objective of the analysis was to identify how the studied variables can lead to an improved waste management in the context of sustainable development at European level, given that the integrated and sustainable waste management capacity must be a top priority and overcome technical issues to embrace the various vital elements of sustainability [11].

## 2. Literature Review

According to Directive 2008/98/EC of the European Parliament, waste is "any substance or object which the holder discards or intends or is required to discard". The Directive establishes and regulates the ways of reducing or preventing possible adverse impacts from waste generation and management [12]. However, since it is necessary to increase the sustainability of the waste management process by promoting the principles of the circular economy, this directive has been modified by the adoption of Directive 851/2018. It aims at adopting additional measures on sustainable production and consumption, covering the whole life cycle of products and setting new, more ambitious, objectives for European countries to move towards a circular economy [13]. A correct and effective approach to waste management can lead to progress in meeting the Millennium Development Goals [14].

There is a lot of research highlighting the aggravation of the problem of generated waste, a problem that threatens the very human-environment relationship itself. A solution to this issue, provided by the studies, is to set up a rigorous waste management plan as an important tool for developing environmental policies by different countries, in order to reduce greenhouse gas emissions [15] towards a transition to a more circular economy [16,17]. One component of this plan is to predict as much as possible the quantities of municipal waste generated [18]. The authors of this study have used multiple regression models applied to panel data on 38 urban and rural localities in the New York area to estimate the amount of residential municipal solid waste, based on independent variables in the climatological, demographic, socio-economic, cost and distance dimensions.

Kolekara et al. [19] have shown some negative aspects in the efficient and accurate determination of the amount of waste generated, including the lack of sufficient data, especially in developing countries and especially in rural areas, as well as the high degree of uncertainty. They present the main existing models used in estimating the amount of waste generated on the basis of economic, socio-demographic or management factors. Thus, as we will see below, many studies identify as a factor with significant action on waste generation a number of households' characteristics: household size, household income and the level of education. Often the application of the identified models is limited to certain regions and depends on the way the waste flows are selected.

Kawai et al. [20] assessed the confidence level and comparability of the various models used in estimating the quantities of municipal solid waste generated per capita. The level of socio-economic development and the environmental policies of some regions generate differences in the amount of municipal solid waste per capita. A low degree of comparability of per capita waste estimation models may come from the use of definitions of the concept of solid municipal waste, different from country to country. The low efficiency of waste data collection or migration flows between urban and rural areas can also affect the confidence of these models, especially in developing countries.

Another direction on which many specialized studies have been focused is the identification of the determinants, of the main factors affecting the waste generation process. Thus, an important share of the total waste generated by the population is the food waste [21,22]. Chalak et al. [23] analyzed the impact of legislation and economic incentives on the amount of food waste generated by households, based on data covering 44 countries.

Significant variations in the amount of food waste between countries with different levels of development and incomes were noted. The results of the analysis highlighted the higher impact of legislative regulations, policies and strategies in the field on the improvement in the amount of waste generated, compared to fiscal measures.

Significant regional differences in the amount of waste generated per capita are addressed and explained by Saladia [24] in terms of the seasonal population which is normally not taken into account in determining the amount of waste in a given region. The author's analysis reveals that per capita waste is positively correlated with the relative contribution of the tertiary sector to GDP creation and negatively correlated with the population over 64 years. The analysis does not certify that there is a significant correlation between waste generated and per capita income. The demographic factor itself

is, however, an important one in modeling the behavior of the waste generation process [25], the applied model highlighting a direct correlation between the amount of waste and the population density.

Another category of studies addresses different components of waste generation (institutional, residential and commercial) [26–31]. Thus, the analysis by Hockett et al. [32] highlighted that retail sales (including restaurant, grocery and clothing sales) and disposal taxes have a significant impact on waste generation. However, the research did not reveal a significant link between industry, construction, personal income, urbanization and waste generation.

Among the determinants of waste generation, urbanization is the one around which many studies have focused, analyzing its impact on the amount of waste generated [33]. According to Ugwuanyi and Isife [34], the weaknesses of the waste management process are: infrastructure precariousness, legal and political framework, environmental issues management, budgetary constraints, overcapacity, environmental education insufficiency, issues which are—to a large extent—related to the degree of urbanization.

The significant direct impact of the increase in the demographic factor and the rapid development pace on the efficiency of the municipal solid waste management system are addressed by Pai et al. [35]. The results of the study show that an increase in the population leads to an exponential increase in the amount of municipal waste generated.

Another category of studies and research focuses on the importance and impact of education on the waste generation process. Thus, Fredrick et al, Sinthumule and Mkumbuzi, and Al-Khatib, et al. [36–40] identify some of the means of educating and raising awareness among urban communities about the waste management process: involving active organizations in education, NGOs and private companies, public meetings, media use, household head training. Analysis based on a cross-sectional, multistage survey has demonstrated the positive effect of public education on optimizing and streamlining urban waste management, but also draws attention to the insufficiency of education provided in the field of waste separation and organic waste management, which represents more than 50% of the total amount of waste generated in major cities.

Knowledge acquisition in the field of waste recycling, reuse, recovery and composting can be achieved even at younger age at school. Therefore, Rada et al. [3] illustrates the role of education provided to young people in the field of environmental protection, amplified by the example they can offer to their families as a way to optimize household waste management. The study covered educational units at different levels of education (primary, secondary, high school) and analyzed the influence of youth age and typology of educational and informational activities on household behavior in waste management. Research results show that waste production depends on the size of the educational institution (expressed as the number of pupils and teachers), on the types of activities carried out outside the teaching hours, and on the habits of the household members.

Also, research has highlighted the way in which households and their socioeconomic characteristics influence the waste management system [41–48]. Using probit regression models, Handayani et al. [49] showed that the level of education and knowledge gained, as well as the income level have a significant influence on household waste management behavior. At the same time, the authors have demonstrated the existence of regional disparities between urban and rural areas in waste management practices. Households whose members have a higher education level and income and are located in the urban area are more likely to increase the amount of waste generated, but also have a higher probability of managing their waste amount, compared to households in rural areas which, generally, have a lower level of education and income. The positive correlation between the educational level of the household members and the generation of waste within it is in line with the results of some studies, like: Sujauddin et al., Thi Thu Nguyen et al., Fang et al., Jörissen et al. [50–53]. Age and gender also generate significant differences in households' behavior regarding waste management, with women and older people displaying greater responsibility than men and young people. Similar results were also obtained by Li et al. [54] and Limbu [9], which also identified a negative correlation between the educational level of family members and the amount of waste generated within the household. A positive correlation

between household income and the amount of waste generated was identified by Zia et al. [55], who also studied the influence of seasons on waste generation. The conclusion was that this process is more pronounced, more intense in spring and winter [56,57].

The residential area of households has a significant direct influence on waste generation (Skourides et al. [8]; Liao [58]), so changing the household location (rural and urban) leads to a significant increase in the amount of waste generated [9,10]. Other variables positively correlated with the amount of waste generated by households are: household size, family members' availability to spend their leisure time outside the household, consumption pattern and household income. Based on the application of a logistic regression model to the data obtained from a sample of 402 respondents, Afroz et al. [5] concluded that household size and income are significant determinants of the amount of waste generated by households, while the age and education of family members can significantly influence their availability to waste recycling. Emerging from the need to improve negative pressures exerted by human activity on the environment, researchers' studies focused on the differences in the waste management system by the location of the household (urban or rural area). Thus, Han et al. [59] and Marshall and Farahbakhsh [60] study the factors influencing waste generated in rural areas and the peculiarities of this process in developing countries. The explanatory variables included in the study were of economic nature (household income and expenditure, types of energy and fuels used and types of existing industries in the rural area), social (population, education and culture) and natural (temperature, precipitation, humidity, harvesting periods). These factors determine the waste content and consumption patterns and can be used as ways in the optimization and improvement of the waste management system.

Daban Astane and Hajilo [61] carried out a quantitative and qualitative analysis of the waste generated in rural Iranian areas on a sample of 318 households, pointing out that about 70% of the average amount of waste generated per person per day is organic waste, 30% being solid waste. It is noted that there are patterns in the waste spatial distribution, with significant differences in the amount of waste generated in the north and northwest areas, areas with significantly higher rates of waste generation than others. The most important determinants of waste generation are household income, assets, the age and attitude of household members towards environmental issues, while behavior and knowledge related to resource efficiency are significant factors in reducing the amount of waste generated.

## 3. Methodology and Data

Taking into account the objective of the study—identifying the long-term relationship between the amount of waste generated per capita and a number of factors influencing their evolution in 29 European countries during 2013–2017, as well as highlighting the similarities and the differences between them, regarding the indicators analyzed—four explanatory variables were selected. These variables, together with the explained variable, represent the five sets of data on which the study is based (Table 1). The datasets were provided by EUROSTAT and processed with Eviews program.

**Table 1.** List of variables used in the analysis.

| Variable Notation | The Variable | Measurement Unit |
|---|---|---|
| W_POP | Amount of waste per capita | t/inhabitant |
| PSE | Pupils and students—as % of total age population | % |
| TS_POP | Classroom teachers and academic staff | people/100 inhabitants |
| POP_C | Population total median equivalised income—Cities | % |
| POP_R | Population total median equivalised income—Rural Areas | % |

Source: Authors' selection, based on EUROSTAT data.

The methodology applied in the study started with the analysis of the main characteristics of the variables included, as well as the identification of the correlation matrix between them.

In analyzing the dynamics of economic processes and systems, frequently occur situations in which the endogenous variables are placed both on the left and right side of the equations of the respective economic models. In these cases a possibility of analysis is the use of Vector Autoregressive Models (VAR), as well as Vector Error Correction (VEC).

Given the complexity of the interdependencies between the variables included in the analysis and the necessity of highlighting the cointegration relationships between them, starting from the matrix $V \in R^{k \times n}$ (where $k$ represents the number of endogenous variables and $n$—the lengths of the data series), the following model were generated: VAR models—for the analysis of the interdependent time series and the effects of perturbations on the values of the variables involved, and VEC models—for identifying the cointegration relationships between the variables.

The general form of a VAR model is:

$$y_t = \sum_{i=1}^{p} A_i \cdot y_{t-i} + B \cdot x_t + \varepsilon_t \tag{1}$$

where $y_t$ is a vector of $k$ endogenous variables, $x_t$ is a vector of $m$ exogenous variables, and $A_i \in R^{k \times p}$, $B \in R^{k \times m}$ are coefficients matrices to be estimated.

VEC models are derived from VAR models and are designed to restrict the long-term behavior of endogenous variables by incorporating cointegration relationships in VEC models, so as to converge to them by allowing short-term adjustments. The term "*cointegration*" is called "Error Correction Term" (ECT), as it allows estimating and correcting short-term deviations from the long-term equilibrium of the studied phenomenon.

One of the most commonly used methods for estimating VEC models is Generalised Method of Moments (GMM), the Arellano, M. and Bond, S. (1991) [62] estimator being applied in the analysis of various growth models.

In the research we started from a VEC model whose general form is:

$$\Delta y_t = \beta \cdot ECT_{t-1}^{y} + \sum_{i=1}^{p} A_i \cdot \Delta y_{t-i} + u_t \tag{2}$$

In Equation (2) $\Delta y_t = y_t - y_{t-1}$ is a vector of $k$ endogenous variables, $A_i \in R^{k \times p}$ are coefficients matrices to be estimated and $\beta \in R^k$ is the vector of ECT coefficients.

Taking into account the relation (2) and the data series included in the matrix V, the corresponding VEC model has the following form:

$$\Delta W\_POP_t = \beta_1 \cdot ECT_{t-1} + \sum_{i=1}^{p} \alpha_{1,i}^{W\_POP} \cdot \Delta W\_POP_{t-i} + \sum_{i=1}^{p} \delta_{1,i}^{PSE} \cdot \Delta PSE_{t-i}$$
$$+ \sum_{i=1}^{p} \delta_{1,i}^{TS\_POP} \cdot \Delta TS\_POP_{t-i} + \sum_{i=1}^{p} \phi_{1,i}^{POP\_C} \cdot \Delta POP\_C_{t-i} + \sum_{i=1}^{p} \gamma_{1,i}^{POP\_R} \cdot \Delta POPR_{t-i} + u_{1,t} \tag{3}$$

$$\Delta PSE_t = \beta_2 \cdot ECT_{t-1} + \sum_{i=1}^{p} \alpha_{2,i}^{W\_POP} \cdot \Delta W\_POP_{t-i} + \sum_{i=1}^{p} \delta_{2,i}^{PSE} \cdot \Delta PSE_{t-i}$$
$$+ \sum_{i=1}^{p} \delta_{2,i}^{TS\_POP} \cdot \Delta TS\_POP_{t-i} + \sum_{i=1}^{p} \phi_{2,i}^{POP\_C} \cdot \Delta POP\_C_{t-i} + \sum_{i=1}^{p} \gamma_{2,i}^{POP\_R} \cdot \Delta POPR_{t-i} + u_{2,t} \tag{4}$$

$$\Delta TS\_POP_t = \beta_3 \cdot ECT_{t-1} + \sum_{i=1}^{p} \alpha_{3,i}^{W\_POP} \cdot \Delta W\_POP_{t-i} + \sum_{i=1}^{p} \delta_{3,i}^{PSE} \cdot \Delta PSE_{t-i}$$
$$+ \sum_{i=1}^{p} \delta_{3,i}^{TS\_POP} \cdot \Delta TS\_POP_{t-i} + \sum_{i=1}^{p} \phi_{3,i}^{POP\_C} \cdot \Delta POP\_C_{t-i} + \sum_{i=1}^{p} \gamma_{3,i}^{POP\_R} \cdot \Delta POPR_{t-i} + u_{3,t} \tag{5}$$

$$\Delta POP\_C_t = \beta_4 \cdot ECT_{t-1} + \sum_{i=1}^{p} \alpha_{4,i}^{W\_POP} \cdot \Delta W\_POP_{t-i} + \sum_{i=1}^{p} \delta_{4,i}^{PSE} \cdot \Delta PSE_{t-i}$$
$$+ \sum_{i=1}^{p} \delta_{4,i}^{TS\_POP} \cdot \Delta TS\_POP_{t-i} + \sum_{i=1}^{p} \phi_{4,i}^{POP\_C} \cdot \Delta POP\_C_{t-i} + \sum_{i=1}^{p} \gamma_{4,i}^{POP\_R} \cdot \Delta POPR_{t-i} + u_{4,t} \tag{6}$$

$$\Delta POP\_R_t = \beta_5 \cdot ECT_{t-1} + \sum_{i=1}^{p} \alpha_{5,i}^{W\_POP} \cdot \Delta W\_POP_{t-i} + \sum_{i=1}^{p} \delta_{5,i}^{PSE} \cdot \Delta PSE_{t-i}$$
$$+ \sum_{i=1}^{p} \delta_{5,i}^{TS\_POP} \cdot \Delta TS\_POP_{t-i} + \sum_{i=1}^{p} \phi_{5,i}^{POP\_C} \cdot \Delta POP\_C_{t-i} + \sum_{i=1}^{p} \gamma_{5,i}^{POP\_R} \cdot \Delta POPR_{t-i} + u_{5,t} \tag{7}$$

A first condition in modeling and testing the causal relationship between the five endogenous variables involves testing the stability of the data series. For this, Unit Root Test of Augmented Dickey-Fuller test was used. The Null Hypothesis states that X (the analyzed variable) has a unit root. The validation condition is to reject the Null Hypothesis for 5% significance level ($\alpha = 0.05$).

Testing the existence of cointegration relationships was carried out with the Johansen Cointegration Test [63], (for which the Null Hypothesis states that there are no cointegration relationships). If this second Null Hypothesis is also rejected, the VEC model (or models) is generated and the statistical significance of the parameter values is tested. In (3) the conditions are $\alpha_{1,j}^{W\_POP}, \delta_{1,j}^{PSE}, \mu_{1,j}^{TS\_POP}, \phi_{1,j}^{POP\_C}, \gamma_{1,j}^{POP\_R} \neq 0$, $(\forall)j \neq 0$ and they are similar to the conditions for Equations (4)–(7).

A second important condition for validating the model is to ensure the system convergence to balance. For this, the ECT coefficient should be negative ($\beta_1 < 0$) and statistically significant for the significance level chosen. For this, the t-statistics test was used.

After identifying the general form of the ECT, its parameters have to be statistically significant. This condition is also checked using the t-statistics test. Confidence level used was 95% ($\alpha = 0.05$). In some situations, a 90% confidence level ($\alpha = 0.10$) was also accepted.

For checking the stability conditions, autoregressive roots graph, as well as the model response to unit impulses applied to its variables are displayed and analyzed.

The methodology used throughout the paper produced results that have led to a clearer picture of the impact of the urbanization process and institutional education on past, present and future evolution of waste amount, in the countries included in the analysis.

## 4. Results

Achieving the objective of finding a way in which the management of the analyzed variables can lead to sustainable waste management and address sustainable European development, requires an approach based on the main features of the variables included in the analysis, as well as the correlation matrix, presented in Table 2.

The results of partial correlations show that the W_POP variable is correlated with the other variables (all correlation coefficients being statistically significant), although the correlation between W_POP and TS_POP is poor. Taking this into account, we will consider W_POP as the main variable in the analysis performed. At the same time, it can be inferred that the PSE, TS_POP and POP_C variables can be considered as independent variables in the first phase, which can influence W_POP. As for the POP_R variable, although the results may indicate a possible endogenousness between POP_C and POP_C, it can be solved by generating the VEC model.

Using VEC models requires prior determination of the integration order. The results obtained by applying the Augmented Dickey-Fuller (ADF) test are presented in Table 3. These lead to the conclusion that series are 1$^{st}$ order integrated and suitable for further cointegration testing.

**Table 2.** Main characteristics of the analyzed variables and the correlation matrix between them.

|  | W_POP | PSE | TS_POP | POP_C | POP_R |
|---|---|---|---|---|---|
| Mean | 0.407152 | 21.37928 | 1.757995 | 39.28739 | 32.77387 |
| Median | 0.412586 | 20.30000 | 1.838117 | 36.10000 | 34.70000 |
| Maximum | 1.182873 | 31.40000 | 2.870871 | 90.40000 | 56.20000 |
| Minimum | 0.191658 | 17.50000 | 0.551509 | 13.00000 | 0.100000 |
| Std. Dev. | 0.138738 | 2.988497 | 0.522968 | 14.10623 | 13.05356 |
| Observations | 112 | 112 | 112 | 112 | 112 |
| W_POP | 1.000000 |  |  |  |  |
|  | - |  |  |  |  |
| PSE | 0.464211 | 1.000000 |  |  |  |
|  | (0.0000) | - |  |  |  |
| TS_POP | 0.189715 | 0.228380 | 1.000000 |  |  |
|  | (0.0461) | (0.0004) | - |  |  |
| POP_C | 0.321884 | 0.057580 | 0.032029 | 1.000000 |  |
|  | (0.0027) | (0.5483) | (0.7386) | - |  |
| POP_R | −0.378688 | −0.191441 | −0.018805 | −0.677338 | 1.000000 |
|  | (0.0000) | (0.0145) | (0.8447) | (0.0000) | - |

**Table 3.** Augmented Dickey-Fuller test statistics.

| D(W_POP) | | D(PSE) | | D(TS_POP) | | D(POP_C) | | D(POP_R) | |
|---|---|---|---|---|---|---|---|---|---|
| t-Stat | Prob. | t-Stat | Prob. | t-Stat | Prob. | t-Stat | Prob. | t-Stat | Prob. |
| −10.83 | 0.000 | −11.18 | 0.000 | −7.86 | 0.000 | −10.43 | 0.000 | −10.22 | 0.000 |
| Test critical values: −3.490772 ***; −2.887909 **; −2.580908 * | | | | | | | | | |

*** 1% level, ** 5% level, * 10% level

In order to determine the co-integration order of VAR (p)—type process between the series in the level, there have been used the information criteria of Akaike (AIC), Schwarz (SC) and Hannan-Quinn (HQ), as well as Final Prediction Error and sequential modified LR statistical test (LR). All the other criteria (Table 4) suggest the order of the VAR process q = 1. Taking into account that the stationarity of the series was obtained by a 1$^{st}$ order differentiation, the chosen process will be VAR (2).

**Table 4.** Results regarding the establishment of the order in the VAR process.

| VAR Lag Order Selection Criteria | | | | | | |
|---|---|---|---|---|---|---|
| Endogenous variables: W_POP PSE TS_POP POP_C POP_R | | | | | | |
| Lag | LogL | LR | FPE | AIC | SC | HQ |
| 0 | −926.1369 | NA | 224.4159 | 19.60288 | 19.73730 | 19.65720 |
| 1 | −770.6526 | 291.3284 * | 14.40106 * | 16.85585 * | 17.66233 * | 17.18173 * |
| 2 | −754.1369 | 29.20688 | 17.29282 | 17.03446 | 18.51302 | 17.63191 |
| 3 | −746.1053 | 13.35776 | 24.99076 | 17.39169 | 19.54232 | 18.26071 |
| * indicates lag order selected by the criterion | | | | | | |
| Each test at 5% level | | | | | | |

Considering the results obtained above and using Johansen's (1988) approach, the existence of three cointegration equations (Table 5) for the chosen significance level was highlighted. At the same time, however, the Max-Eigen Statistic values for 5% significance level indicate the existence of a single regression equation, which includes the five variables considered as endogenous ones.

Taking into account the fact that the Max-Eigen Statistic is more restrictive, the intention was to identify the model that co-integrates all five variables included in the analysis. Testing their endogeneity

was performed with the Pairwise Granger causality test (Null Hypothesis: the variables analyzed cannot be exogenous variables in the model) and their belonging to the model was verified using Lag Excluding Tests (Null Hypothesis: the analyzed variable is not excluded from the model).

**Table 5.** The results of Cointegration Rank Test.

| Unrestricted Cointegration Rank Test (Trace) | | | | | Unrestricted Cointegration Rank Test (Maximum) | | | |
|---|---|---|---|---|---|---|---|---|
| Hypoth. No. of CE(s) | Eigen Value | Trace Statistic | 0.05 Critical Value | Prob.** | Hypoth. No. of CE(s) | Max-Eigen Statistic | 0.05 Critical Value | Prob.** |
| None * | 0.2821 | 98.2859 | 76.9727 | 0.0005 | None * | 34.78528 | 34.6905 | 0.0489 |
| At most 1 * | 0.1938 | 63.5006 | 54.0790 | 0.0058 | At most 1 | 22.62919 | 28.5881 | 0.2391 |
| At most 2 * | 0.1836 | 40.8715 | 35.1927 | 0.0110 | At most 2 | 21.29950 | 22.2996 | 0.0685 |
| At most 3 | 0.1132 | 19.5720 | 20.2618 | 0.0620 | At most 3 | 12.62394 | 15.8921 | 0.1526 |
| At most 4 | 0.0640 | 6.94806 | 9.16454 | 0.1292 | At most 4 | 6.948063 | 9.16454 | 0.1292 |
| Lags interval (in first differences): 1 to 2 | | | | | * denotes rejection of the hypothesis at the 0.05 level | | | |
| Trace test indicates 3 cointegrating eqn(s) at the 0.05 level | | | | | ** MacKinnon-Haug-Michelis (1999) *p*-values | | | |
| Max-eigenvalue test indicates 1 cointegrating eqn(s) at the 0.05 level | | | | | | | | |

The results of Pairwise Granger causality (Table 6) lead to accepting the Null Hypothesis for all five variables. Thus, in the case of the dependent variable D(W_POP) that reflects the change in the volume of waste per capita per time unit (year), as a result of the change in one time unit of the Pupils and Students as % of total age population, Classroom teachers and academic staff, Population total median equivalised income—Cities, Population total median equivalised income—Rural Areas, the individual probabilities corresponding to the values of the $\chi 2$ statistic (Chi-square) and the value corresponding to the model as a whole (Prob = 0.7894) are higher than the 5% significance level ($\alpha$ = 0.05). The same conclusion is reached for the other four variables.

**Table 6.** The results of VEC Granger Causality test.

| Dependent variable: D(W_POP) | | | Dependent variable: D(PSE) | | | df |
|---|---|---|---|---|---|---|
| Excluded | Chi-sq | Prob. | Excluded | Chi-sq | Prob. | |
| D(PSE) | 1.1881 | 0.5521 | D(W_POP) | 0.7669 | 0.6815 | 2 |
| D(TS_POP) | 0.2389 | 0.8874 | D(TS_POP) | 1.0993 | 0.5771 | 2 |
| D(POP_C) | 3.4708 | 0.1763 | D(POP_C) | 0.2063 | 0.9020 | 2 |
| D(POP_R) | 2.0618 | 0.3567 | D(POP_R) | 0.1836 | 0.9123 | 2 |
| All | 4.6969 | 0.7894 | All | 2.6027 | 0.9568 | 8 |

| Dependent variable: D(TS_POP) | | | Dependent variable: D(POP_C) | | | Dependent variable: D(POP_R) | | | df |
|---|---|---|---|---|---|---|---|---|---|
| Excluded | Chi-sq | Prob. | Excluded | Chi-sq | Prob. | Excluded | Chi-sq | Prob. | |
| D(W_POP) | 1.2353 | 0.5392 | D(W_POP) | 0.1720 | 0.9176 | D(W_POP) | 0.6387 | 0.7266 | 2 |
| D(PSE) | 4.5637 | 0.1021 | D(PSE) | 0.7143 | 0.6996 | D(PSE) | 0.2923 | 0.8640 | 2 |
| D(POP_C) | 3.6037 | 0.1650 | D(TS_POP) | 0.8618 | 0.6499 | D(TS_POP) | 0.5196 | 0.7712 | 2 |
| D(POP_R) | 3.0890 | 0.2134 | D(POP_R) | 1.5575 | 0.4590 | D(POP_C) | 0.5200 | 0.7710 | 2 |
| All | 10.048 | 0.2616 | All | 3.3028 | 0.9139 | All | 1.6624 | 0.9897 | 8 |

The above tests highlighted the existence of a model in which all five variables are considered endogenous. To exclude or include them from the VEC model, the critical value of the Chi-squared statistic, for 5% significance level ($\alpha$ = 0.05) and 5 degrees of freedom whose value is 11.070498 was taken into account.

The results of the Lag Exclusion Wald Tests (Table 7) show that all Chi-squared test statistics for both DLag_1 and DLag_2 are strictly lower than the critical value (11.0705). Consequently, the Null Hypothesis is accepted. It follows that none of the variables will be excluded from the model. The same conclusion is reached by analyzing *p_value* values, which are significantly higher than the chosen significance level.

**Table 7.** VEC Lag Exclusion Wald Tests.

|  | D(W_POP) | D(PSE) | D(TS_POP) | D(POP_C) | D(POP_R) | Joint |
|---|---|---|---|---|---|---|
| DLag 1 | 4.625024 | 1.008971 | 8.533598 | 0.975558 | 0.488963 | 20.94764 |
|  | [0.463332] | [0.961839] | [0.129177] | [0.964513] | [0.992522] | [0.695526] |
| DLag 2 | 2.139070 | 1.068857 | 5.832351 | 1.777931 | 2.437144 | 13.91055 |
|  | [0.829587] | [0.956823] | [0.322875] | [0.878939] | [0.785930] | [0.963269] |
| df | 5 | 5 | 5 | 5 | 5 | 25 |

Numbers in [ ] are *p*-values

Taking into account the obtained results, the VEC model, which includes all five analyzed variables, was generated (Table 8). Testing the model validity and the statistical significance of its parameters was performed with the Student (t) test. In the case of this model, for the significance level chosen, the critical test value is 1.98238. Since the value of CointEq1 constant (coefficient $\beta_1$ in Equation (3)) is negative ($-0.007086$) and statistically significant (t-statistics = $-2.33718 > $ t-critic = 1.98238), the VEC model is statistically significant. Also, since all the t-statistic values of the model coefficients are higher than the critical value, it follows that the Null Hypothesis is rejected and the Alternative Hypothesis is accepted. All coefficients are statistically significant.

**Table 8.** The estimation of VEC coefficients and their Standard errors and t-statistics values.

| Cointegrating Eq: | CointEq1 | Standard Errors | t-Statistics |
|---|---|---|---|
| W_POP(-1) | 1.000000 |  |  |
| PSE(-1) | 0.648967 | (0.15372) | [4.22178] |
| TS_POP(-1) | 4.076959 | (0.81624) | [4.99479] |
| POP_C(-1) | −0.135140 | (0.03740) | [−3.61348] |
| POP_R(-1) | −0.135135 | (0.03988) | [−3.38871] |
| C | 15.93563 | (4.50194) | [3.53972] |
| Error Correction: | D(W_POP) |  |  |
| CointEq1 | −0.007086 | (0.00530) | [−2.33718] |

Regarding the stability of the VEC model, this is highlighted on the one hand by how the analyzed model reacted to the application of a unit impulse on its variables, and on the other hand that all the roots of the characteristic polynomial are included in the unit circle (Figure 1).

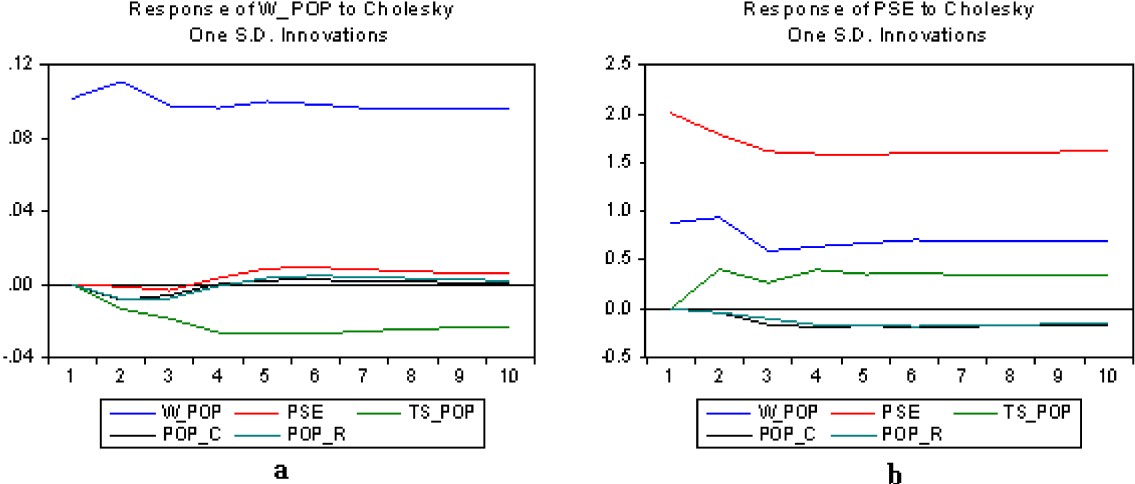

**Figure 1.** *Cont.*

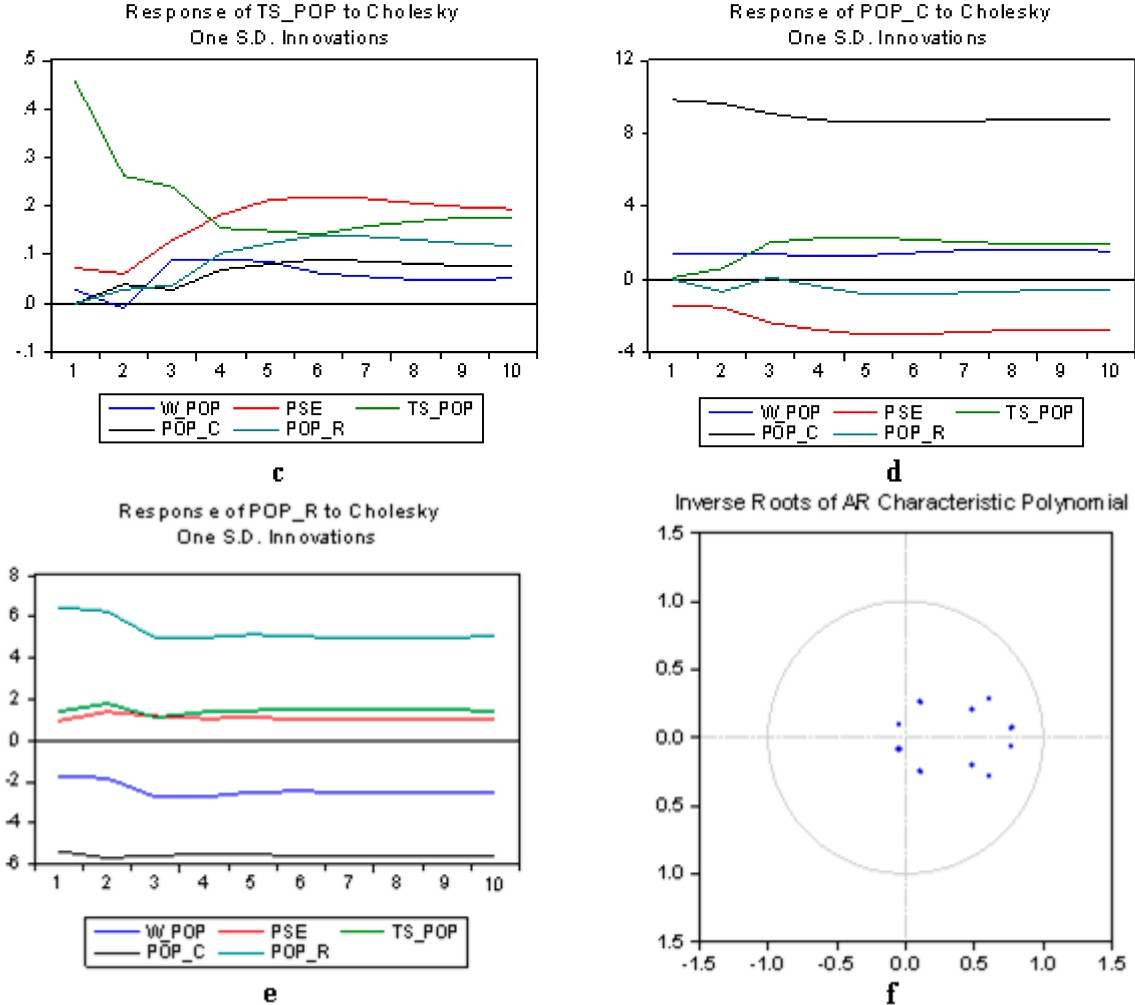

**Figure 1.** Response of W_POP (**a**), PSE (**b**), TS_POP (**c**), POP_C (**d**), POP_R (**e**) to Cholesky One S.D. Innovations, and Inverse Roots of AR characteristic polynomial (**f**).

The long-term relation, corresponding to the Error Correction Term (ECT$_{t+1}$), based on Equations (3)–(7) has the following form:

$$
\begin{aligned}
d(W\_POP) = \quad & -0.007 \cdot (W\_POP_{t-1} + 0.649 \cdot PSE_{t-1} + 4.077 \cdot TS\_POP_{t-1} - 0.135 \cdot POP\_C_{t-1} \\
& -0.135 \cdot POP\_R_{t-1} + 19.936 ) + 0.107 \cdot W\_POP_{t-1} - 0.142 \cdot W\_POP_{t-2} - 0.005 \cdot PSE_{t-1} \\
& -0.005 \cdot PSE_{t-2} + 0.004 \cdot TS\_POP_{t-1} + 0.012 \cdot TS\_POP_{t-2} - 0.003 \cdot POP\_C_{t-1} \\
& -0.0005 \cdot POP\_C_{t-2} - 0.002 \cdot POP\_R_{t-1} - 0.0009 \cdot POP\_R_{t-2}
\end{aligned}
\tag{8}
$$

The long-term relationships of Error Correction Term (ECT1), based on Equations (3)–(7) is:

$$
\begin{aligned}
W\_POP = \quad & -0.6489 \cdot PSE - 4.0769 \cdot TS\_POP \\
& +0.1351 \cdot POP\_C + 0.1351 \cdot POP\_R - 19.9356
\end{aligned}
\tag{9}
$$

The regression coefficients show the long-lasting reverse connection between the amount of waste and Pupils and students—as % of the total age population, respectively Classroom teachers and academic staff per 100 inhabitants, meaning that an increase in the education level leads to a reduction in the waste amount. As for the other two factors (Population total median equivalised income—Cities, Population total median equivalised income—Rural Areas), their influence on the amount of waste is direct, without a significant long-term difference between urban and rural

areas. It follows that the education level is a primordial factor, which can act favorably to sustainable development.

## 5. Discussions

Considering that, in time, limitations exist regarding the complete and detailed analysis of the amount of waste generated, mainly due to difficulties in data collection [19], the paper addressed the issue of waste quantity behavior, by studying the interdependence level of two categories of factors: educational and demographic, by degree of urbanization, at the level of 29 European countries.

In this context, an autoregressive vector model, followed by a Vector Error Correction model have been applied, on 2013–2017 data, referring to pupils and students, classroom teachers and academic staff by educational level and the population by degree of urbanization (cities, rural areas).

Analyzing the long-term relationship between the amount of waste, education and the demographic factor by residence area, it is pointed out that the amount of waste is reduced as the level of education increases and is increased with the demographic factor, taking into account the existence of insignificant difference by the two types of residence areas.

Regarding the first factor (the educational one), it is noted that at institutional level sustainable development can be addressed through the upgrading of the educational waste recycling-oriented process, in order to improve the protection and the quality of the environment, through increasing the competence of the services provided, resulted in well-trained graduates, both theoretically and practically.

Currently, more and more employers are looking for graduates with sustainable education. They need to know the aspects of sustainability and to have the skills to use them in their professional work.

At both institutional and social levels, universities can be seen as having an important role in the community. At the same time, students can demonstrate they are the most important agents of change, both in the environment and in society, that they have environmental, social and economic knowledge about sustainability and that they have a new system of values, motivation and other abilities to produce change.

Regarding the second factor (the demographic one, by residence area) it is noted that in many low-urbanization areas there is no uniform, equal distribution of solid waste collection services, but there is a hierarchy of rural localities according to their financial strength to provide solid residential waste collection services, as well as some informal features of governance (such as involving local wealthy families in local governance) [64,65].

The solid municipal waste recycling system in developing countries highlights that sustainable solid waste management is a concern for the mankind as a whole and is also addressed in the United Nations Millennium Development Goals, as a way of reversing the negative impact of human activity on environment [66]. Therefore, adequate management of solid waste should be addressed in order to reduce poverty, reduce child mortality and improve the population health. While large-scale sustainable solid waste management methods are used in developed countries, it is advisable in developing countries to take over and use these methods on an increasing scale.

The system may include an economic and financial mechanism that respects the general principles, especially "the polluter pays" principle and the subsidiarity principle. In addition, the integrated waste management system should also consider promoting information and awareness system for all actors involved in the action, as well as a modern system for obtaining complete and accurate data and information, adequate to national and European reporting requirements.

## 6. Conclusions

A major challenge for sustainable development is how to encourage activities that positively influence the environment and discourage those which cause environmental damage [67]. As waste is

an important component of the environmental pressure exerted by humans, the capacity to manage the activities of waste amount reduction is a priority task in managerial strategies.

In this context, in order to capture future waste behaviour (as a measure of sustainability), starting from the 2013–2017 data, two factorial plans were addressed: education (pupils and students by education level and classroom teachers and academic staff by education level) and demographic, by degree of urbanization (cities, rural areas). The research was conducted at the level of 29 EU countries. The innovative contribution of the article consists of combining the educational and demographic factors by residential areas, as well as the statistical indicators selected for the quantification of the two factors. At the same time, originality also consists in the fact that the applied model analyzes the long-term implication of these factors on the quantity of waste generated, a less discussed aspect in other specialized studies.

Applying the Vector Autoregressive Model and Vector Error Correction Model led to a long-term relationship between waste quantity, education and demographic factor by residence. At the same time, this relationship shows the reduction of the waste quantity is significantly influenced by the increase of the educational level, while the demographic factor, due to the insignificant difference between the two types of residence environments, determines the increase of the waste amount.

In this context, it is possible to highlight the way in which we can manipulate each factor so that to encourage an improvement in the waste management process.

The educational factor puts the spotlight on the educational institutions. Thus, they must have the capacity to allocate resources intelligently on the one hand to become sustainable, and on the other hand to provide learners (pupils, students) with a living experience in a sustainable environment. At the same time, they must be the catalyst for the changes that are expected and demanded at the society level as a whole, but also by their graduates in order to become tools for the society transformation towards sustainable development.

As a conclusion, it can be said that both pupils and students must have many of the attributes needed to act as agents of change and need to be trained as active citizens to boost and sustain the global economy.

Regarding the demographic factor by urbanization degree, the comparative analysis of the data shows the waste management system in low and middle income countries is less developed in rural areas than in urban areas, a fact also confirmed by other studies in the literature [68,69]. The main problems of the waste management system in rural areas in less developed countries are: lack of political and financial tools that can be applied to improve waste management, lack of long-term vision of waste management policy, lack of elementary measures, which have long been applied in developed countries, lack of funds, lack of environmental responsibility, poor quality of waste management services [70]. There are significant differences in the coverage of waste collection services between major cities and rural areas in less developed countries [71], while in developed countries these services cover entirely the rural areas.

In this context, it is necessary to act in the future on the main factors with a significant influence on the waste management system: the existence of legislative regulations in the field, the application of incentive schemes, the cost of waste management procedures, the budgets allocated for this purpose, education and knowledge of the importance of waste collection, household income, availability of local population to pay waste collection and evacuation taxes, existence of markets based on recycled materials, involvement of companies and small businesses in the waste management process. It is also recommended to conduct awareness and accountability programs for urban population, focused on the need to reduce the amount of waste, by periodically evacuating and recycling the waste generated.

Designing and applying innovative solutions that should be both resource efficient and waste management systems add value to business and contribute to sustainability. In this way, we can talk about an improvement in operations management and environmental management, that can be integrated at operational level and include the waste management supply chain.

The implementation of an integrated waste management system requires the adaptation and development of an institutional and organizational framework, for the introduction of national requirements and harmonization with European structures.

The authors intend to extend the analysis by identifying other variables with significant impact on the evolution of the amount of waste generated, as well as forecasting it for a future period, by considering other typologies of econometric models. As the model is limited to European countries, the authors wish to analyze the pattern of the relationship between the amount of waste generated, the level of education and the environment of residence also in non-European countries.

**Author Contributions:** All the authors were equally involved in the documentation phase, in choosing the research methodology, in data analysis, as well as in results analysis and in discussions. All the authors have equally participated to the manuscript preparation and have approved the submitted manuscript.

**Funding:** This research received no external funding.

**Conflicts of Interest:** The authors declare no conflict of interest.

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
