# Peer review of "The Impact of Education and Residential Environment on Long-Term Waste Management Behavior in the Context of Sustainability"

_sustainability, doi:10.3390/su11143775_

Reviewer 1 Report

The proposed manuscript aims to assess "The impact of education and residential environment on long-term waste management behavior in the context of sustainability". The topic is of great interest and relevance. The methodological framework is linear and does not have particular problems or omissions allowing you to make everything work strictly from a scientific perspective, and easier to read. 

There are however some minor aspects that need to be still fixed.

Although the paper deals with an interesting topic, few substantial revisions are required. The paper is satisfactory written, but needs a careful editing. Further, while the study aim and background are well presented, few repetitions occurring in the paper should be avoided. Accordingly, it is opinion of this reviewer to accept with minor revisions the proposed manuscript for a publication on this journal.

Therefore, firstly author/s should deeply and critically discuss the study findings posing attention on the innovativeness of the study. Secondly, since the study lack of clear implications, author/s should fully outline them in a separate section of the conclusion paragraph.

The introduction and, above all, the conclusions can be improved in order to show better aim and results for further studies in the topic.

Reviewer 2 Report

The research addresses an issue of interest in waste management by trying to find direct links between education, residential environment and waste management in the context of sustainability. The study is well structured and contains a rich research of literature in the field. The methodology and results are properly explained.

Hereinafter, I have a number of suggestions for improvement that I would like to share with you:

Point 1.           Line 33: the keywords must be separated all by semicolon (;);

Point 2.           Line 52: you have an extra point after “markets”;

Point 3.           Line 96: before and after "/" please do not use spaces;

Point 4.           Line 110: please observe the multiple reference formatting (no space after comma);

Point 5.           Please pay attention to the formatting of titles and table titles in the paper (Eg: at R241 the table title must end with a point; before and after "/" please do not use spaces; R243 must be removed); also Line 259 is not necessary. Idem Line 282, 282, 316, 319, 323, 327, 334, 337, 342, 345, 360, 363, 373, 377, 386, 389, 395, 399;

Point 6.           Line 261: the text following an equation need not be a new paragraph. Idem Eq (2);

Point 7.           Line 298: Missing the end point of the sentence;

Point 8.           In Results, if you specify with which program you perform the analysis, it is no longer necessary to specify the source for each table and figure;

Point 9.           Line 335: Missing table ending border;

Point 10.       The reviewer suggests that for Figure 1, which has multiple panels, to be used for each, notation from (a) to (f) according to MDPI template;

Point 11.       Line 483: The format for the paragraph is wrong;

Point 12.       Please review and prepare the reference list according to the template requirements.
